# Jointly Learning Sentence Embeddings and Syntax with Unsupervised Tree-LSTMs

## Abstract

We introduce a neural network that represents sentences by composing their words according to induced binary parse trees. We use Tree-LSTM as our composition function, applied along a tree structure found by a fully differentiable natural language chart parser. Our model simultaneously optimises both the composition function and the parser, thus eliminating the need for externally-provided parse trees which are normally required for Tree-LSTM. It can therefore be seen as a tree-based RNN that is unsupervised with respect to the parse trees. As it is fully differentiable, our model is easily trained with an off-the-shelf gradient descent method and backpropagation. We demonstrate that it achieves better performance compared to various supervised Tree-LSTM architectures on a textual entailment task and a reverse dictionary task. Finally, we show how performance can be improved with an attention mechanism which fully exploits the parse chart, by attending over all possible subspans of the sentence.

## 1 Introduction

Recurrent neural networks, in particular the Long Short-Term Memory (LSTM) architecture (Hochreiter & Schmidhuber, 1997) and some of its variants (Graves & Schmidhuber, 2005; Bahdanau et al., 2014), have been widely applied to problems in natural language processing. Examples include language modelling (Sundermeyer et al., 2012; Józefowicz et al., 2016), textual entailment (Bowman et al., 2015; Sha et al., 2016), and machine translation (Bahdanau et al., 2014; Sutskever et al., 2014) amongst others.

The topology of an LSTM network is linear: words are read sequentially, normally in left-to-right order. However, language is known to have an underlying hierarchical, tree-like structure (Chomsky, 1957). How to capture this structure in a neural network, and whether doing so leads to improved performance on common linguistic tasks, is an open question. The Tree-LSTM network (Tai et al., 2015; Zhu et al., 2015) provides a possible answer, by generalising the LSTM to tree-structured topologies. It was shown to be more effective than a standard LSTM in semantic relatedness and sentiment analysis tasks.

Despite their superior performance on these tasks, Tree-LSTM networks have the drawback of requiring an extra labelling of the input sentences in the form of parse trees. These can be either provided by an automatic parser (Tai et al., 2015), or taken from a gold-standard resource such as the Penn Treebank (Kiperwasser & Goldberg, 2016). Yogatama et al. (2016) proposed to remove this requirement by including a shift-reduce parser in the model, to be optimised alongside the composition function based on a downstream task. This makes the full model non-differentiable so it needs to be trained with reinforcement learning, which can be slow due to high variance.

Our proposed approach is to include a fully differentiable chart parser in the model, inspired by the CYK constituency parser (Cocke, 1969; Younger, 1967; Kasami, 1965). Due to the parser being differentiable, the entire model can be trained end-to-end for a downstream task by using stochastic gradient descent. Our model is also unsupervised with respect to the parse trees, similar to Yogatama et al. (2016). We show that the proposed method outperforms baseline Tree-LSTM architectures based on fully left-branching, right-branching, and supervised parse trees on a textual entailment task and a reverse dictionary task. We also introduce an attention mechanism in the spirit of Bahdanau et al. (2014) for our model, which attends over all possible subspans of the source sentence via the parse chart.

## 2 RELATED WORK

Our work can be seen as part of a wider class of sentence embedding models that let their composition order be guided by a tree structure. These can be further split into two groups: (1) models that rely on traditional syntactic parse trees, usually provided as input, and (2) models that induce a tree structure based on some downstream task.

In the first group, Paperno et al. (2014) take inspiration from the standard Montagovian semantic treatment of composition. They model nouns as vectors, and relational words that take arguments (such as adjectives, that combine with nouns) as tensors, with tensor contraction representing application (Coecke et al., 2011). These tensors are trained via linear regression based on a downstream task, but the tree that determines their order of application is expected to be provided as input. Socher et al. (2012) and Socher et al. (2013) also rely on external trees, but use recursive neural networks as the composition function.

Instead of using a single parse tree, Le & Zuidema (2015) propose a model that takes as input a parse forest from an external parser, in order to deal with uncertainty. The authors use a convolutional neural network composition function and, like our model, rely on a mechanism similar to the one employed by the CYK parser to process the trees. Ma et al. (2015) propose a related model, also making use of syntactic information and convolutional networks to obtain a representation in a bottom-up manner. Convolutional neural networks can also be used to produce embeddings without the use of tree structures, such as in Kalchbrenner et al. (2014).

Bowman et al. (2016) propose an RNN that produces sentence embeddings optimised for a down-stream task, with a composition function that works similarly to a shift-reduce parser. The model is able to operate on unparsed data by using an integrated parser. However, it is trained to mimic the decisions that would be taken by an external parser, and is therefore not free to explore using different tree structures. Dyer et al. (2016) introduce a probabilistic model of sentences that explicitly models nested, hierarchical relationships among words and phrases. They too rely on a shift-reduce parsing mechanism to obtain trees, trained on a corpus of gold-standard trees.

In the second group, Yogatama et al. (2016) shows the most similarities to our proposed model. The authors use reinforcement learning to learn tree structures for a neural network model similar to Bowman et al. (2016), taking performance on a downstream task that uses the computed sentence representations as the reward signal. Kim et al. (2017) take a slightly different approach: they formalise a dependency parser as a graphical model, viewed as an extension to attention mechanisms, and hand-optimise the backpropagation step through the inference algorithm.

## 3 MODELS

All the models take a sentence as input, represented as an ordered sequence of words. Each word $w_i \in \mathcal{V}$ in the vocabulary is encoded as a (learned) word embedding $\boldsymbol{w}_i \in \mathbb{R}^d$. The models then output a sentence representation $\boldsymbol{h} \in \mathbb{R}^D$, where the output space $\mathbb{R}^D$ does not necessarily coincide with the input space $\mathbb{R}^d$.

### 3.1 BAG OF WORDS

Our simplest baseline is a bag-of-words (BoW) model. Due to its reliance on addition, which is commutative, any information on the original order of words is lost. Given a sentence encoded by embeddings $\boldsymbol{w}_1, \ldots, \boldsymbol{w}_n$ it computes

$$\boldsymbol{h} = \sum_{i=1}^{n} \tanh\left(\mathbf{W}\boldsymbol{w}_i + \boldsymbol{b}\right),$$

where $\mathbf{W}$ is a learned input projection matrix.

## 3.2 LSTM

An obvious choice for a baseline is the popular Long Short-Term Memory (LSTM) architecture of Hochreiter & Schmidhuber (1997). It is a recurrent neural network that, given a sentence encoded by embeddings $\boldsymbol{w}_1, \ldots, \boldsymbol{w}_T$, runs for $T$ time steps $t = 1 \ldots T$ and computes

$$\begin{bmatrix} \boldsymbol{i}_t \\ \boldsymbol{f}_t \\ \boldsymbol{u}_t \\ \boldsymbol{o}_t \end{bmatrix} = \mathbf{W}\boldsymbol{w}_t + \mathbf{U}\boldsymbol{h}_{t-1} + \boldsymbol{b},$$

$$\boldsymbol{c}_t = \boldsymbol{c}_{t-1} \odot \sigma(\boldsymbol{f}_t) + \tanh(\boldsymbol{u}_t) \odot \sigma(\boldsymbol{i}_t),$$
$$\boldsymbol{h}_t = \sigma(\boldsymbol{o}_t) \odot \tanh(\boldsymbol{c}_t),$$

where $\sigma(x) = \frac{1}{1+e^{-x}}$ is the standard logistic function. The LSTM is parametrised by the matrices $\mathbf{W} \in \mathbb{R}^{4D \times d}$, $\mathbf{U} \in \mathbb{R}^{4D \times D}$, and the bias vector $\boldsymbol{b} \in \mathbb{R}^{4D}$. The vectors $\sigma(\boldsymbol{i}_t), \sigma(\boldsymbol{f}_t), \sigma(\boldsymbol{o}_t) \in \mathbb{R}^D$ are known as *input*, *forget*, and *output* gates respectively, while we call the vector $\tanh(\boldsymbol{u}_t)$ the *candidate update*. We take $\boldsymbol{h}_T$, the $\boldsymbol{h}$-state of the last time step, as the final representation of the sentence.

Following the recommendation of Jozefowicz et al. (2015), we deviate slightly from the vanilla LSTM architecture described above by also adding a bias of 1 to the forget gate, which was found to improve performance.

## 3.3 TREE-LSTM

Tree-LSTMs are a family of extensions of the LSTM architecture to tree structures (Tai et al., 2015; Zhu et al., 2015). We implement the version designed for binary constituency trees. Given a node with children labelled $L$ and $R$, its representation is computed as

$$\begin{bmatrix} \boldsymbol{i} \\ \boldsymbol{f}_L \\ \boldsymbol{f}_R \\ \boldsymbol{u} \\ \boldsymbol{o} \end{bmatrix} = \mathbf{W}\boldsymbol{w} + \mathbf{U}\boldsymbol{h}_L + \mathbf{V}\boldsymbol{h}_R + \boldsymbol{b}, \tag{1}$$

$$\boldsymbol{c} = \boldsymbol{c}_L \odot \sigma(\boldsymbol{f}_L) + \boldsymbol{c}_R \odot \sigma(\boldsymbol{f}_R) + \tanh(\boldsymbol{u}) \odot \sigma(\boldsymbol{i}), \tag{2}$$
$$\boldsymbol{h} = \sigma(\boldsymbol{o}) \odot \tanh(\boldsymbol{c}), \tag{3}$$

where $\boldsymbol{w}$ in (1) is a word embedding, only nonzero at the leaves of the parse tree; and $\boldsymbol{h}_L, \boldsymbol{h}_R$ and $\boldsymbol{c}_L, \boldsymbol{c}_R$ are the node children's $\boldsymbol{h}$- and $\boldsymbol{c}$-states, only nonzero at the branches. These computations are repeated recursively following the tree structure, and the representation of the whole sentence is given by the $\boldsymbol{h}$-state of the root node. Analogously to our LSTM implementation, here we also add a bias of 1 to the forget gates.

## 3.4 UNSUPERVISED TREE-LSTM

While the Tree-LSTM is very powerful, it requires as input not only the sentence, but also a parse tree structure defined over it. Our proposed extension optimises this step away, by including a basic CYK-style (Cocke, 1969; Younger, 1967; Kasami, 1965) chart parser in the model. The parser has the property of being fully differentiable, and can therefore be trained jointly with the Tree-LSTM composition function for some downstream task.

Table 1: Chart for the sentence "neuro linguistic programming rocks".

| | | | neuro linguistic programming rocks |
| --- | --- | --- | --- |
| | | neuro linguistic programming | linguistic programming rocks |
| | neuro linguistic | linguistic programming | programming rocks |
| neuro | linguistic | programming | rocks |

The CYK parser relies on a *chart* data structure, which provides a convenient way of representing the possible binary parse trees of a sentence, according to some grammar. Here we use the chart as an efficient means to store all possible binary-branching trees, effectively using a grammar with only a single non-terminal. This is sketched in simplified form in Table 1 for an example input. The chart is drawn as a diagonal matrix, where the bottom row contains the individual words of the input sentence. The $n^{\text{th}}$ row contains all cells with branch nodes spanning $n$ words (here each cell is represented simply by the span – see Figure 1 below for a forest representation of the nodes in all possible trees). By combining nodes in this chart in various ways it is possible to efficiently represent every binary parse tree of the input sentence.

The unsupervised Tree-LSTM uses an analogous chart to guide the order of composition. Instead of storing sets of non-terminals, however, as in a standard chart parser, here each cell is made up of a pair of vectors $(\boldsymbol{h}, \boldsymbol{c})$ representing the state of the Tree-LSTM RNN at that particular node in the tree. The process starts at the bottom row, where each cell is filled in by calculating the Tree-LSTM output (1)-(3) with $\boldsymbol{w}$ set to the embedding of the corresponding word. These are the leaves of the parse tree. Then, the second row is computed by repeatedly calling the Tree-LSTM with the appropriate children. This row contains the nodes that are directly combining two leaves. They might not all be needed for the final parse tree: some leaves might connect directly to higher-level nodes, which have not yet been considered. However, they are all computed, as we cannot yet know whether there are better ways of connecting them to the tree. This decision is made at a later stage.

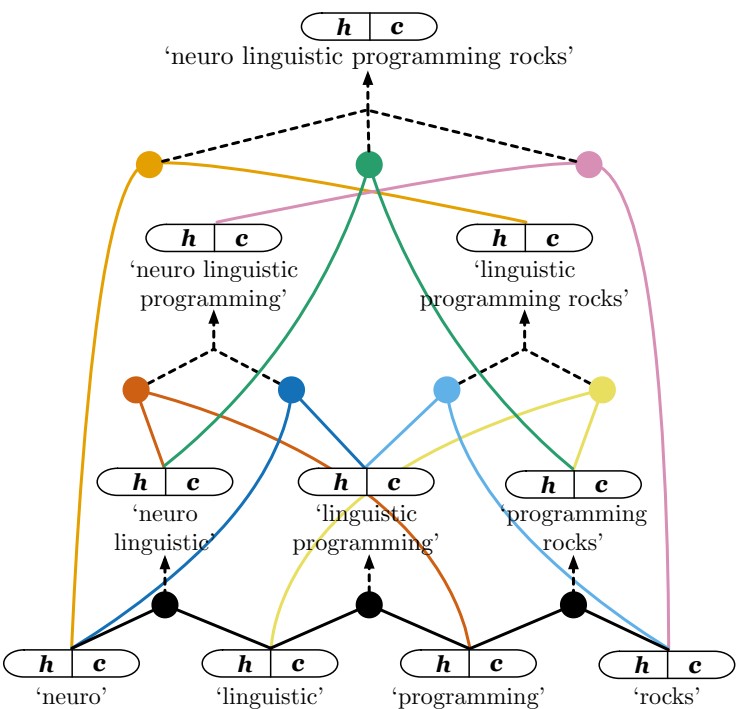

Figure 1: Unsupervised Tree-LSTM network structure for the sentence "neuro linguistic programming rocks".

Starting from the third row, ambiguity arises since constituents can be built up in more than one way: for example, the constituent "neuro linguistic programming" in Table 1 can be made up either by combining the leaf "neuro" and the second-row node "linguistic programming", or by combining the second-row node "neuro linguistic" and the leaf "programming". In these cases, all possible compositions are performed, leading to a set of candidate constituents $(\boldsymbol{c}_1, \boldsymbol{h}_2), \dots, (\boldsymbol{c}_n, \boldsymbol{h}_n)$. Each is assigned an energy, given by

$$e_i = \cos(\boldsymbol{u}, \boldsymbol{h}_i), \tag{4}$$

where $\cos(\cdot, \cdot)$ indicates the cosine similarity function and $\boldsymbol{u}$ is a (trained) vector of weights. All energies are then passed through a softmax function to normalise them, and the cell representation is

finally calculated as a weighted sum of all candidates using the softmax output:

$$s_i = \text{softmax}(e_i/t), \tag{5}$$

$$\boldsymbol{c} = \sum_{i=1}^{n} s_i \boldsymbol{c}_i, \qquad \boldsymbol{h} = \sum_{i=1}^{n} s_i \boldsymbol{h}_i.$$

The softmax uses a temperature hyperparameter $t$ which, for small values, has the effect of making the distribution sparse by making the highest score tend to 1. In all our experiments the temperature is initialised as $t = 1$, and is smoothly decreasing as $t = 1/2^e$, where $e \in \mathbb{Q}$ is the fraction of training epochs that have been completed. In the limit as $t \to 0^+$, this mechanism will only select the highest scoring option, and is equivalent to the argmax operation. The same procedure is repeated for all higher rows, and the final output is given by the $\boldsymbol{h}$-state of the top cell of the chart.

The whole process is sketched in Figure 1 for an example sentence. Note how, for instance, the final sentence representation can be obtained in three different ways, each represented by a coloured circle. All are computed, and the final representation is a weighted sum of the three, represented by the dotted lines. When the temperature $t$ in (5) reaches very low values, this effectively reduces to the single "best" tree, as selected by gradient descent.

## 4 EXPERIMENTS

All models are implemented in Python 3.5.2 with the DyNet neural network library (Neubig et al., 2017) at commit `25be489`. The code for all following experiments will be made available on the first author's website[1] shortly after the publication date of this article. Performance on the development data is used to determine when to stop training. Each model is trained three times, and the test set performance is reported for the model performing best on the development set.

The textual entailment model was trained on a 2.2 GHz Intel Xeon E5-2660 CPU, and took three days to converge. The reverse dictionary model was trained on a NVIDIA GeForce GTX TITAN Black GPU, and took five days to converge.

In addition to the baselines already described in §3, for the following experiments we also train two additional Tree-LSTM models that use a fixed composition order: one that uses a fully left-branching tree, and one that uses a fully right-branching tree.

### 4.1 TEXTUAL ENTAILMENT

We test our model and baselines on the Stanford Natural Language Inference task (Bowman et al., 2015), consisting of 570 k manually annotated pairs of sentences. Given two sentences, the aim is to predict whether the first *entails*, *contradicts*, or is *neutral* with respect to the second. For example, given "children smiling and waving at camera" and "there are children present", the model would be expected to predict *entailment*.

For this experiment, we choose 100D input embeddings, initialised with 100D GloVe vectors (Pennington et al., 2014) and with out-of-vocabulary words set to the average of all other vectors. This results in a $100 \times 37\,369$ word embedding matrix, fine-tuned during training. For the supervised Tree-LSTM model, we used the parse trees included in the dataset. For training we used the Adam optimisation algorithm (Kingma & Ba, 2014), with a batch size of 16.

Given a pair of sentences, one of the models is used to produce the embeddings $\boldsymbol{s_1}, \boldsymbol{s_2} \in \mathbb{R}^{100}$. Following Yogatama et al. (2016) and Bowman et al. (2016), we then compute

$$\boldsymbol{u} = (\boldsymbol{s_1} - \boldsymbol{s_2})^2,$$
$$\boldsymbol{v} = \boldsymbol{s_1} \odot \boldsymbol{s_2}, \tag{6}$$
$$\boldsymbol{q} = \text{ReLU}\left(\mathbf{A} \begin{bmatrix} \boldsymbol{u} \\ \boldsymbol{v} \\ \boldsymbol{s_1} \\ \boldsymbol{s_2} \end{bmatrix} + \boldsymbol{a}\right),$$

---

[1] https://www.whitehouse.gov/

Table 2: Test set accuracy (higher is better) on the SNLI dataset, and number of parameters. We report separately the number of intrinsic model parameters and the number of word embedding parameters. Other encoding-based models are also reported.

| Model | Test Accuracy | # Parameters |
|---|---|---|
| 100D Bag-of-words | 77.6 % | 91 k + 3.7 M |
| 100D LSTM | 82.2 % | 161 k + 3.7 M |
| 100D Left-branching Tree-LSTM | 82.1 % | 231 k + 3.7 M |
| 100D Right-branching Tree-LSTM | 82.5 % | 231 k + 3.7 M |
| 100D Supervised Tree-LSTM | 82.5 % | 231 k + 3.7 M |
| 100D Unupervised Tree-LSTM | **82.8** % | 231 k + 3.7 M |
| Bowman et al. (2015), 100D LSTM | 77.6 % | 220 k + ? |
| Bowman et al. (2016), 300D SPINN | 83.2 % | 3.7 M + ? |
| Yogatama et al. (2016), 100D latent | 80.5 % | 500 k + 1.8 M |
| Shen et al. (2017), 300D DiSAN | 85.6 % | 2.35 M + ? |

Table 3: Test set accuracy (higher is better) on the SNLI dataset for the two attention models.

| Model | Test Accuracy |
|---|---|
| 100D LSTM + attention | 82.7 % |
| 100D Unupervised Tree-LSTM + attention | **83.2** % |

where $\mathbf{A} \in \mathbb{R}^{200 \times 400}$ and $\boldsymbol{a} \in \mathbb{R}^{200}$ are trained parameters. Finally, the correct label is predicted by $p(\hat{y} = c \mid \boldsymbol{q}; \mathbf{B}, \boldsymbol{b}) \propto \exp(\mathbf{B}_c \boldsymbol{q} + \boldsymbol{b}_c)$, where $\mathbf{B} \in \mathbb{R}^{3 \times 200}$ and $\boldsymbol{b} \in \mathbb{R}^3$ are trained parameters.

Table 2 lists the accuracy and number of parameters for our model, baselines, as well as other sentence embedding models in the literature. When the information is available, we report both the number of intrinsic model parameters as well as the number of word embedding parameters. For other models these figures are based on the data from the SNLI website[2] and the original papers.[3]

### 4.1.1 ATTENTION

Attention is a mechanism which allows a model to soft-search for relevant parts of a sentence. It has been shown to be effective in a variety of linguistic tasks, such as machine translation (Bahdanau et al., 2014; Vaswani et al., 2017), summarisation (Rush et al., 2015), and textual entailment (Shen et al., 2017).

In the spirit of Bahdanau et al. (2014), we modify our LSTM model such that it returns not just the output of the last time step, but rather the outputs for all steps. Thus, we no longer have a single pair of vectors $\boldsymbol{s_1}, \boldsymbol{s_2}$ as in (6), but rather two lists of vectors $\boldsymbol{s_{1,1}}, \ldots, \boldsymbol{s_{1,n_1}}$ and $\boldsymbol{s_{2,1}}, \ldots, \boldsymbol{s_{2,n_2}}$. Then, we replace $\boldsymbol{s_1}$ in (6) with

$$\boldsymbol{s'_1} = \frac{\sum_{i=1}^{n_1} \exp\left(f(\boldsymbol{s_{1,i}}, \boldsymbol{s_{2,n_2}})\right) \boldsymbol{s_{1,i}}}{\sum_{j=1}^{n_1} \exp\left(f(\boldsymbol{s_{1,j}}, \boldsymbol{s_{2,n_2}})\right)}, \qquad \text{with } f(\boldsymbol{x}, \boldsymbol{y}) \equiv \boldsymbol{a} \cdot \tanh\left(\mathbf{A}_i \boldsymbol{x} + \mathbf{A}_s \boldsymbol{y}\right),$$

where $f$ is the *attention mechanism*, with vector parameter $\boldsymbol{a}$ and matrix parameters $\mathbf{A}_i, \mathbf{A}_s$. This can be interpreted as attending over sentence 1, informed by the context of sentence 2 via the vector $\boldsymbol{s_{2,n_2}}$. Similarly, $\boldsymbol{s_2}$ is replaced by an analogously defined $\boldsymbol{s'_2}$, with separate attention parameters.

We also extend the mechanism of Bahdanau et al. (2014) to the Unsupervised Tree-LSTM. In this case, instead of attending over the list of outputs of an LSTM at different time steps, attention is over the whole chart structure described in §3.4. Thus, the model is no longer attending over all *words* in the source sentences, but rather over all their possible *subspans*. The results for both attention-augmented models are reported in Table 3.

---

[2] https://nlp.stanford.edu/projects/snli/

[3] The number of word embedding parameters in the model of Yogatama et al. (2016) is lower than ours. This is due to Yogatama et al. (2016) filtering out infrequent words. One of the authors reported (personal communication) that using the full vocabulary did not change their result significantly.

Table 4: Median rank (lower is better) and accuracies (higher is better) at 10 and 100 on the three test sets for the reverse dictionary task: seen words (S), unseen words (U), and concept descriptions (C).

| Model | Median rank | | | Top 10 accuracy | | | Top 100 accuracy | | |
|---|---|---|---|---|---|---|---|---|---|
| | S | U | C | S | U | C | S | U | C |
| Bag-of-words | 75.0 | 66.0 | 70.5 | 30.3% | 29.9% | 25.8% | 53.7% | 55.2% | 56.6% |
| LSTM | **57.5** | 59.0 | 48.5 | 28.9% | 29.7% | 29.3% | 55.3% | 56.8% | 57.1% |
| Left-branching Tree-LSTM | 78.0 | 64.0 | 48.0 | 28.9% | 28.3% | 28.8% | 52.7% | 54.8% | 61.1% |
| Right-branching Tree-LSTM | 70.5 | 51.0 | 42.5 | 30.1% | 30.9% | 29.8% | 54.5% | **58.0%** | 62.1% |
| Supervised Tree-LSTM | 108.5 | 79.0 | 160.5 | 23.1% | 26.9% | 20.2% | 49.0% | 52.9% | 42.4% |
| Unsupervised Tree-LSTM | 58.5 | **40.0** | **40.0** | **30.9%** | **33.4%** | **30.3%** | **56.1%** | 57.1% | **62.6%** |
| Hill et al. (2016) 512D LSTM | 19 | 19 | 26 | 44% | 44% | 38% | 70% | 69% | 66% |
| Hill et al. (2016) 500D BoW | 15 | 14 | 28 | 46% | 46% | 36% | 71% | 71% | 66% |

## 4.2 REVERSE DICTIONARY

We also test our model and baselines on the reverse dictionary task of Hill et al. (2016), which consists of 852 k word-definition pairs. The aim is to retrieve the name of a concept from a list of words, given its definition. For example, when provided with the sentence "control consisting of a mechanical device for controlling fluid flow", a model would be expected to rank the word "valve" above other confounders in a list. We use three test sets provided by the authors: two sets involving word definitions, either seen during training or held out; and one set involving concept descriptions instead of formal definitions. Performance is measured via three statistics: the *median rank* of the correct answer over a list of over 66 k words; and the proportion of cases in which the correct answer appears in the top 10 and 100 ranked words (*top 10 accuracy* and *top 100 accuracy*).

As output embeddings, we use the 500D CBOW vectors (Mikolov et al., 2013) provided by the authors. As input embeddings we use the same vectors, reduced to 256 dimensions with PCA. Given a training definition as a sequence of (input) embeddings $\boldsymbol{w}_1, \ldots, \boldsymbol{w}_n \in \mathbb{R}^{256}$, the model produces an embedding $\boldsymbol{s} \in \mathbb{R}^{256}$ which is then mapped to the output space via a trained projection matrix $\mathbf{W} \in \mathbb{R}^{500 \times 256}$. The training objective to be maximised is then the cosine similarity $\cos(\mathbf{W}\boldsymbol{s}, \boldsymbol{d})$ between the definition embedding and the (output) embedding $\boldsymbol{d}$ of the word being defined. For the supervised Tree-LSTM model, we additionally parsed the definitions with Stanford CoreNLP (Manning et al., 2014) to obtain parse trees.

We use simple stochastic gradient descent for training. The first 128 batches are held out from the training set to be used as development data. The softmax temperature in (5) is allowed to decrease as described in §3.4 until it reaches a value of 0.005, and then kept constant. This was found to have the best performance on the development set.

Table 4 shows the results for our model and baselines, as well as the numbers for the cosine-based "w2v" models of Hill et al. (2016), taken directly from their paper.[4] Our bag-of-words model consists of 193.8 k parameters; our LSTM uses 653 k parameters; the fixed-branching, supervised, and unsupervised Tree-LSTM models all use 1.1 M parameters. On top of these, the input word embeddings consist of $113\,123 \times 256$ parameters. Output embeddings are not counted as they are not updated during training.

## 5 DISCUSSION

The results in Tables 2-4 show a strong performance of the Unsupervised Tree-LSTM against our tested baselines, as well as other similar methods in the literature with a comparable number of parameters.

---

[4]We note that our initial reimplementation of the "w2v cosine" models of Hill et al. (2016), using vectors supplied by the authors, achieved a slightly different set of results than theirs. We include their numbers for completeness. Our own baselines are architecturally different from those of Hill et al. (2016), but we found our variants to perform better on development data.

For the textual entailment task, our model outperforms all baselines including the supervised Tree-LSTM, as well as some of the other sentence embedding models in the literature with a higher number of parameters. The use of attention, extended for the Unsupervised Tree-LSTM to be over all possible subspans, further improves performance.

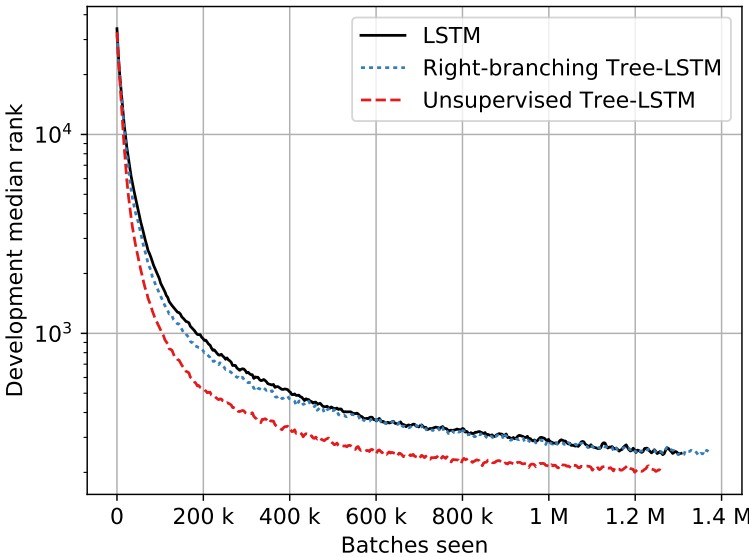

Figure 2: Median rank (lower is better) on the development set for the reverse dictionary task.

In the reverse dictionary task, the poor performance of the supervised Tree-LSTM can be explained by the unusual tokenisation used in the dataset of Hill et al. (2016): punctuation is simply stripped, turning e.g. "(archaic) a section of a poem" into "archaic a section of a poem", or stripping away the semicolons in long lists of synonyms. On the one hand, this might seem unfair on the supervised Tree-LSTM, which received suboptimal trees as input. On the other hand, it demonstrates the robustness of our method to noisy data. Our model also performed well in comparison to the LSTM and the other Tree-LSTM baselines. Despite the slower training time due to the additional complexity, Figure 2 shows how our model needed fewer training examples to reach convergence in this task.

Following Yogatama et al. (2016), we also manually inspect the learned trees to see how closely they match conventional syntax trees, as would typically be assigned by trained linguists. We analyse the same four sentences they chose. The trees produced by our model are shown in Figure 3. One notable feature is the fact that verbs are joined with their subject noun phrases first, which differs from the standard verb phrase structure. However, formalisms such as combinatory categorial grammar (Steedman, 2000), through type-raising and composition operators, do allow such constituents. The spans of prepositional phrases in (b), (c) and (d) are correctly identified at the highest level; but only in (d) does the structure of the subtree match convention. As could be expected, other features such as the attachment of the full stops or of some determiners do not appear to match human intuition.

## 6 CONCLUSIONS

We presented a fully differentiable model to jointly learn sentence embeddings and syntax, based on the Tree-LSTM composition function. We demonstrated its benefits over standard Tree-LSTM on a textual entailment task and a reverse dictionary task. Introducing an attention mechanism over the parse chart was shown to further improve performance for the textual entailment task. The model is conceptually simple, and easy to train via backpropagation and stochastic gradient descent with popular deep learning toolkits based on dynamic computation graphs such as DyNet (Neubig et al., 2017) and PyTorch.[5]

---

[5] https://github.com/pytorch/pytorch

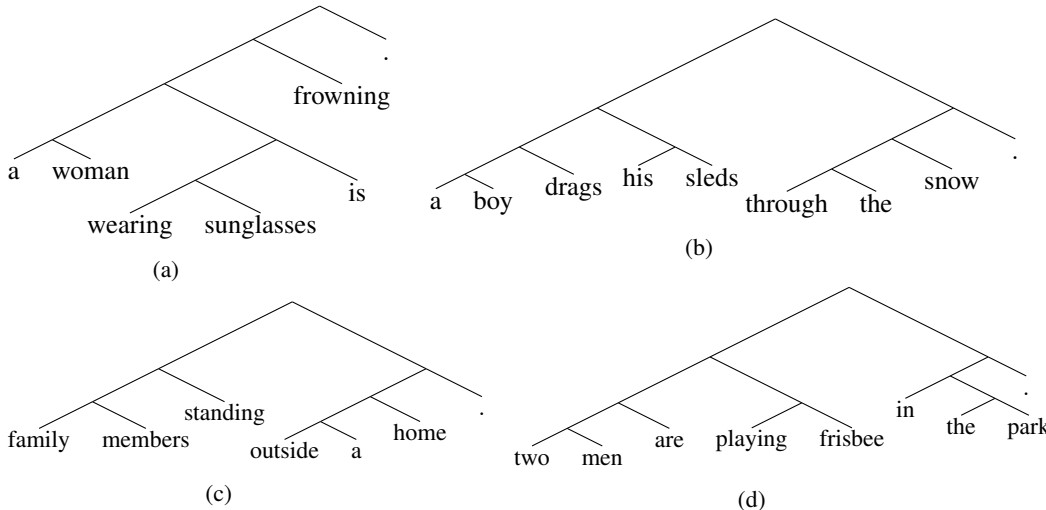

Figure 3: Binary parse trees of sentences from the SNLI dataset induced by the Unsupervised Tree-LSTM model.

The unsupervised Tree-LSTM we presented is relatively simple, but could be plausibly improved by combining it with aspects of other models. It should be noted in particular that (4), the function assigning an energy to alternative ways of forming constituents, is extremely basic and does not rely on any global information on the sentence. Using a more complex function, perhaps relying on a mechanism such as the tracking LSTM in Bowman et al. (2016), might lead to improvements in performance. Techniques such as batch normalization (Ioffe & Szegedy, 2015) or layer normalization (Ba et al., 2016) might also lead to further improvements.

In future work, it may be possible to obtain trees closer to human intuition by training models to perform well on multiple tasks instead of a single one, an important feature for intelligent agents to demonstrate (Legg & Hutter, 2007). Elastic weight consolidation (Kirkpatrick et al., 2017) has been shown to help with multitask learning, and could be readily applied to our model.

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
