# OpenReview forum: "Jointly Learning Sentence Embeddings and Syntax with Unsupervised Tree-LSTMs"
_ICLR.cc/2018/Conference — Reject_

### Official Review · AnonReviewer1 · 2017-11-28

**Rating:** 5
**Confidence:** 4

**Review:**

This paper proposes to jointly learning a semantic objective and inducing a binary tree structure for word composition, which is similar to (Yogatama et al, 2017). Differently from (Yogatama et al, 2017), this paper doesn’t use reinforcement learning to induce a hard structure, but adopts a chart parser manner and basically learns all the possible binary parse trees in a soft way.

Overall, I think it is really an interesting direction and the proposed method sounds reasonable. However, I am concerned about the following points:

- The improvements are really limited on both the SNLI and the Reverse Dictionary tasks. (Yogatama et al, 2017) demonstrate results on 5 tasks and I think it’d be helpful to present results on a diverse set of tasks and see if conclusions can generally hold. Also, it would be much better to have a direct comparison to (Yogatama et al, 2017), including the performance and also the induced tree structures.

- The computational complexity of this model shouldn’t be neglected. If I understand it correctly, the model needs to compute O(N^3) LSTM compositions. This should be at least discussed in the paper. And I am not also sure how hard this model is being converged in all experiments (compared to LSTM or supervised tree-LSTM).

- I am wondering about the effects of the temperature parameter t. Is that important for training?

Minor:
- What is the difference between LSTM and left-branching LSTM?
- I am not sure if the attention overt chart is a highlight of the paper or not. If so, better move that part to the models section instead of mention it briefly in the experiments section. Also, if any visualization (over the chart) can be provided, that’d be helpful to understand what is going on.

---

### Official Review · AnonReviewer3 · 2017-11-28

**Rating:** 6
**Confidence:** 4

**Review:**

Summary: The paper proposes to use the CYK chart-based mechanism to compute vector representations for sentences in a bottom-up manner as in recursive NNs. The key idea is to maintain a chart to take into account all possible spans. The paper also introduces an attention method over chart cells. The experimental results show that the propped model outperforms tree-lstm using external parsers.

Comment: I kinda like the idea of using chart, and the attention over chart cells. The paper is very well written.
- My only concern about the novelty of the paper is that the idea of using CYK chart-based mechanism is already explored in Le and Zuidema (2015).
- Le and Zudema use pooling and this paper uses weighted sum. Any differences in terms of theory and experiment?
- I like the new attention over chart cells. But I was surprised that the authors didn’t use it in the second experiment (reverse dictionary).
- In table 2, it is difficult for me to see if the difference between unsupervised tree-lstm and right-branching tree-lstm (0.3%) is “good enough”. In which cases the former did correctly but the latter didn’t?
- In table 3, what if we use the right-branching tree-lstm with attention?
- In table 4, why do Hill et al lstm and bow perform much better than the others?

---

### Official Review · AnonReviewer2 · 2017-12-01
**LSTM-version of Le and Zuidema (2015)**

**Rating:** 4
**Confidence:** 4

**Review:**

The paper presents a model titled the "unsupervised tree-LSTM," in which the authors mash up a dynamic-programming chart and a recurrent neural network. As far as I can glean, the topology of the neural network is constructed using the chart of a CKY parser. When combining different constituents, an energy function is computed (equation 6) and the resulting energies are passed through a softmax. The architecture achieves impressive results on two tasks: SNLI and the reverse dictionary of Hill et al. (2016).

Overall, I found the paper deeply uninspired. The authors downplay the similarity of their paper to that of Le and Zuidema (2015), which  I did not appreciate. It's true that Le and Zuidema take a parse forest from an existing parser, but it still contains an exponential number of trees, as does the work in here. Note that exposition in Le and Zuidema (2015) discusses the pruned case as well, i.e., a compete parse forest. The authors of this paper simply write "Le and Zuidema (2015) propose a model that takes as input a parse forest from an external parser, in order to deal with uncertainty." I would encourage the authors to revisit Le and Zuidema (2015), especially section 3.2, and consider the technical innovations over the existing work. I believe the primary difference (other using an LSTM instead of a convnet) is to replace max-pooling with softmax-pooling. Do these two architectural changes matter? The experiments offer no empirical comparison. In short, the insight of having an end-to-end differentiable function based on a dynamic-programming chart is pretty common -- the idea is in the air. The authors provide yet another instantiation of such an approach, but this time with an LSTM.

The technical exposition is also relatively poor. The authors could have expressed their network using a clean recursion, following the parse chart, but opted not to, and, instead,  provided a round-about explanation in English. Thus, despite the strong results, I would not like to see this work in the proceedings, due to the lack of originality and poor technical discussion. If the paper were substantially cleaned-up, I would be willing to increase my rating.

---

### Decision · Program_Chairs · 2018-01-29
**ICLR 2018 Conference Acceptance Decision**

**Decision:**

Reject

**Comment:**

Though the general direction is interesting and relevant to ICLR, the novelty is limited. As reviewers point out it is very similar to Le & Zuidema (2015), with few modifications (using LSTM word representations, a different type of pooling). However, it is not clear if they are necessary  as there is no direct comparison (e.g., using a different type of pooling). Overall, though the submission is generally solid,  it does not seem appropriate for ICLR.

+ solid
+ well written
- novelty limited
- relation to Le & Zuidema is underplayed